# Effects of Different Heavy Metal Stressors on the Endophytic Community Composition and Diversity of *Symphytum officinale*

**DOI:** 10.3390/microorganisms12030477

**Published:** 2024-02-27

**Authors:** Jing Ma, Dawei Chen, Yifan Xu, Yue Liu, Lele Liu, Jing Huang, Ruochun Gao, Jie Bai, Qinzheng Hou

**Affiliations:** College of Life Science, Northwest Normal University, Lanzhou 730070, China; mjing1121@163.com (J.M.); gansudaweichen@126.com (D.C.); xuyifan0825@126.com (Y.X.); 15636312319@163.com (Y.L.); yybearhao123@126.com (L.L.); 18893787680@163.com (J.H.); 18215143109@163.com (R.G.); 18298807303@163.com (J.B.)

**Keywords:** endophytes, diversity, correlation analysis, co-occurrence network

## Abstract

Endophytes play an important role in helping plants resist heavy metal stress. However, little is known about the effects of different heavy metals on the diversity and composition of endophyte communities. In this study, we used 16S and ITS amplicon sequencing to reveal the structure and function of endophytes in *Symphytum officinale* under different heavy metal stressors. The results showed that the endophytic fungal diversity decreased compared with the control under the different heavy metals stressors, while the diversity of endophytic bacteria showed an increasing trend. The biomarker analysis indicated that Zn and Pb stress led to obvious branches. Specific OTUs analysis showed that there were 1224, 597, and 1004 OTUs specific under Zn, Pb, and Cd stress in the bacterial community and 135, 81, and 110 OTUs specific under Zn, Pb, and Cd stress in the fungal community. The co-occurrence network showed changes in microbial interactions under heavy metal contamination conditions, suggesting that endophytic bacteria play an important role in the resistance of host plants. The Spearman analysis showed that the correlation between endophytic bacteria and endophytic fungi in relation to heavy metal transport exhibited variations. Our results expand the knowledge of the relationships of plant–microbe interactions and offer pivotal information to reveal the role of endophytes under different heavy metal stress conditions.

## 1. Introduction

Plant endophytes are microorganisms residing within the internal spaces of plant organs, such as roots, stems, leaves, and seeds. Importantly, they typically do not induce diseases and often establish symbiotic relationships with the host plant [1]. Certain endophytes confer benefits to host plants, including the promotion of plant growth and assistance in enhancing the host plant’s resilience to adverse external environments or serving as potential biocontrol resources [2]. Notably, halotolerant and halophilic bacteria have the capability to form symbiotic associations with halophytes. Genera, such as Bacillus, Neorhizobium, Delftia, Pseudomonas, Cupriavidus, and Stenotrophomonas, have been reported for their ability to enhance rice growth and reduce the accumulation of heavy metals (HMs) in grains. Additionally, Bacillus, Paenibacillus, Klebsiella, Herbaspirillum, and Pseudomonas have been identified among the endophytes, displaying multiple plant growth-promoting (PGP) activities, such as phosphate solubilization and the production of phytohormones, siderophores, and ACC deaminase [3]. These activities underscore their beneficial roles in improving rice growth under stressful conditions. Given the widespread pollution of heavy metals due to rapid industrial development and the extensive use of chemical fertilizers and pesticides [4], analyzing the diversity and composition of endophytes in plants becomes crucial for enhancing plant stress resistance. Ascomycota is present in a high proportion of endophytic fungi. Most of the reported strains with HM tolerance are members of Ascomycota, such as Aspergillus, Penicillium, Fusarium, this is consistent with the results of this study. In addition, some studies have shown that in HM-treated soils, the bacterial community structure undergoes significant recombination [5]. Therefore, it is of great significance to explore the changes in endophytic bacteria under heavy metal stress to improve plant resistance to heavy metal stress [6]. Therefore, explicating endophyte resources has the potential to improve plant resistance to heavy metals and repair heavy metal pollution [7].

*S. officinale* has significant tolerance to a variety of heavy metals, and we speculate this phenomenon may be related to endophytes. Thus, we used amplicon sequencing to explore the bacterial and fungal community composition and structure of endophytes in *S. officinale* under different heavy metal stressors. These results may lay a foundation for expanding the knowledge on plant–microbe relationships under heavy metal stress. It is important to strengthen plant resistance to heavy metal stress [8].

## 2. Experimental Material

In order to compare the community structure and diversity of endophytes in *S. officinale* under different heavy metal stressors, the present experiment was conducted by artificially cultivating the field-collected roots of *S. officinale* under artificial stress. The study was carried out in 2022 in Yunding Mountain, A’gan Town, Lanzhou City, Gansu Province (altitude: 1800 m, N: 35°54′23.59″ E: 103°52′39.05″). July and August are the time of year when *S. officinale* grows to maturity, usually with a more mature root system. Therefore, the roots were dug out and brought back to the laboratory for preservation and used for potting in the experiments at a later stage.

### 2.1. S. officinale Cultivation and Sampling

Sandy soil was taken and sieved to remove the sandy soil that was too fine or too coarse. This was mixed with vermiculite 2:1, sterilized at a high temperature (121 °C, 20 min), and filled in pots (caliber: 18 × 21). The soaked roots were packed into pots and placed in a greenhouse with a temperature of 25 ± 1 °C during the daytime and 20 ± 1 °C during the nighttime, with a relative humidity of 65~75%. A set of 10 replicates ensured that the germination rate reached 90%, and the seedlings grew until 1~2 leaves were grown in approximately 7 days. We selected uniformly growing *S. officinale* for the experiments and three metal ions, Zn^2+^, Pb^2+^, and Cd^2+^, with chemical reagents corresponding to ZnCl_2_, PbCl_2_, and CdCl_2_ (produced by Tianjin Best Chemical Co., LTD., In Tianjin, China) at concentrations of 1200 mg/L, 600 mg/L, and 500 mg/L, respectively. A total of 700 mL of corresponding metal solutions were poured each time in intervals of 5 days. The control group had the same dose of tap water added each time. When the treatment group appeared to be different from the control group, the roots of polymeric grass were taken to determine the structure of its internal microbial community.

In order to compare the diversity of endophytic bacteria in *S. officinale* under three types of heavy metal stress conditions, the root tissues were collected and tested for *S. officinale* subjected to Zn^+2^, Pb^+2^, and Cd^+2^ heavy metal stress. Three biological replicates were obtained for each heavy metal ion stress condition. The different tissue sample types were separated and washed with tap water to remove the surface sediments. Subsequently, they were rinsed three times with distilled water, followed by sequential immersion in 75% ethanol for 5 min, 2.5% NaClO for 2 min, and again in 75% ethanol for 1 min. After several rinses with sterile water to eliminate the residual ethanol and sodium hypochlorite on the surface, the sterile water from the final rinse was applied to potato agar medium (PDA) and nutrient agar medium (NA) [9]. The plates were then incubated at 28 °C for ten days to assess the effectiveness of surface disinfection. All the sterilized samples were stored in a refrigerator at −80 °C until further processing.

### 2.2. DNA Extraction and High-Throughput Sequencing

A 0.5 g sample of *S. officinale* roots was meticulously washed and subjected to triple surface sterilization. Subsequently, endophytic bacterial DNA was extracted from the *S. officinale* roots following a meticulous protocol in triplicates, adhering to the manufacturer’s guidelines. The DNA concentration was determined using a NanoDrop spectrophotometer (Model 2000; Thermo Fisher Scientific, Waltham, MA, USA) and stored at −20 °C for subsequent PCR analysis. For the amplification of the bacterial 16S rRNA gene V5-V7 region, specific primers 5′AACMGGATTAGATACCCKG and 3′ACGTCATCCCCACCTTCC were employed. Simultaneously, the fungal ITS1-1F region was amplified using primers 5′-CTTGGTCATTTAGAGGAAGTAA and 3′ GCTGCGTTCTTCATCGATGC. The sequencing procedure was conducted on the Illumina MiSeq platform utilizing a paired-end protocol. All PCR reactions were carried out with Phusion^®^ High—Fidelity PCR Master Mix (New England Biolabs, In Beijing, China), 0.2 µM of forward and reverse primers, and approximately 10 ng of template DNA. The thermal cycling consisted of initial denaturation at 98C for 1 min followed by 30 cycles of denaturation at 98 °C for 10 s, annealing at 50 °C for 30 s, and elongation at 72 °C for 30 s and 72 °C for 5 min. We mixed the same volume of 1X loading buffer (containing SYB green) with the PCR products and conducted electrophoresis on a 2% agarose gel for detection. The PCR products were mixed in equidensity ratios. Then, the mixture of PCR products was purified with a Universal DNA Purification Kit (TianGen, In Beijing China, Catalog #: DP214). Sequencing libraries were generated using NEB Next^®^ Ultra™ II FS DNA PCR-free Library Prep Kit (New England Biolabs, New England, USA, Catalog #: E7430L) following the manufacturer’s recommendations, and the indices were added. The library was checked with Qubit and real-time PCR for quantification and a bioanalyzer for size distribution detection. The quantified libraries were pooled and sequenced on Illumina platforms according to the effective library concentration and data amount required [10]. The raw sequences underwent quality filtering using USEARCH (v10.0) following previously established protocols. Subsequently, the bacterial and fungal sequences were taxonomically classified employing the SILVA (v13.2) and UNITE (v8.0) databases, respectively. In total, 496,822 high-quality bacterial reads and 229,607 fungal reads were obtained from the 12 samples, and these were further categorized into 4006 bacterial and 725 fungal operational taxonomic units (OTUs). The bacterial OTUs were compared with the data set obtained with FAPROTAX (script version 1.1), and for the output functional table, we used the default settings. The fungal functional profiles were inferred using PICRUSt2 and the program FUNGuild, where FUNGuild v1.0 was used to determine the functional group of the fungi. FUNGuild v1.0 is a flat database hosted by GitHub [11].

### 2.3. Data Analysis

Paired-end reads were assigned to the samples based on their unique barcode and truncated by cutting off the barcode and primer sequence. The paired-end reads were merged using FLASH (V1.2. 1 1, http://ccb.jhu.edu/software/FLASH/ accessed on 21 November 2023), a very fast and accurate analysis tool, which was designed to merge paired-end reads when at least some of the reads overlap the read generated from the opposite end of the same DNA fragment, and the splicing sequences were called raw tags. Quality filtering on the raw tags was performed using fastp (Version 0.23.1) software to obtain high-quality clean tags. The tags were compared with the reference database (Silva database (16S/18S), https://www.arb-silva.de/ (accessed on 21 November 2023); Unite database (ITS), https://unite.ut.ee/ accessed on 21 November 2023) using the UCHIME algorithm (http://www.drive5.com/usearch/manual/uchime_algo.html accessed on 21 November 2023) to detect the chimera sequences, and then the chimera sequences were removed. Then, the effective tags were finally obtained. Species annotation was performed using QIIME2((Version QIIME2-202006)) software. For 16S, the annotation database was the Silva database, while for ITS, it was the Unite database. In order to study the phylogenetic relationship of each ASV and the differences of the dominant species among the different samples (groups), multiple sequence alignment was performed using QIIME2 software. The species richness and diversity of the microbial community were estimated using a series of statistical analysis indices in the alpha diversity analysis. The Chao1 index was used to calculate the abundance of microorganisms in the downed log samples, while the Shannon entropy, Simpson, and Pielou evenness indices were used to measure the microorganism diversity (richness and evenness). The diversity index was calculated and analyzed using Mothur (version v1.43.0) [11]. The rarefaction curves, composition, and alpha diversity analyses were conducted using https://cloud.majorbio.com/ (accessed on 21 November 2023). A principal coordinates analysis (PCoA) based on the weighted UniFrac distance matrices was performed using the “amplicon” package in R (version 3.6.1). Furthermore, a permutational multivariate analysis of variance (PERMANOVA) was carried out, utilizing the adonis function from the Vegan package in R.

Co-occurrence network analyses were carried out using the “igraph” package in R, with networks visualized through Gephi [12]. The Circos diagram illustrated the variations in the abundance of key microorganisms at the phylum and genus levels, as well as the changes in functional gene abundance under different heavy metal stress conditions in *S. officinale*. A Venn diagram was employed to depict the number of unique and shared microbial species during the various stages of log decomposition. Linear discriminant analysis effect size (LEfSe) biomarker analysis was utilized to explore the dissimilarities in the species composition and functional composition among the samples. A bar chart was used to visualize the alterations in the endophytic community under diverse heavy metal stress conditions. The relationship between microbial communities and physicochemical characteristics of the downed logs was quantified through redundancy analysis (RDA). A heatmap and cluster analysis were applied to demonstrate the distribution and classification of the microbial functions of *S. officinale* under distinct heavy metal stress conditions, while an ANOVA (analysis of variance) and Duncan analysis were conducted to identify the significant differences in the physicochemical characteristics and functions across the groups [13]. Finally, based on the HUMAnN2 analysis results, the responsible species for these functionalities were determined, followed by drawing a column chart comparing the abundance of the functional genes among different samples along with their respective species differences. All the graphics were generated using R 4.0.4 software along with the relevant software packages [14].

## 3. Results

### 3.1. Surface Sterilization Efficiency

Sterile water was used to wash the sediment of the polymer roots, and then sterile paper was used to absorb the water. Then, the samples were soaked in 75% ethanol for 30 s, then in 2% sodium hypochlorite for 30s, followed by washing with sterile water four times, and finally, sterile filter paper was used to absorb the water on the surface [15]. The 100 μL of sterile water used for the previous cleaning of the polymeric roots was coated on nutrient AGAR (NA) solid medium and cultured at 28 °C for 72 h. Following cultivation, the absence of colonies on the NA medium indicated the efficacy of the surface sterilization applied to the samples. This outcome supports the viability of employing the same methodology for the subsequent experiments [16].

### 3.2. Analysis of the Sequencing Data and Alpha Diversity

A total of 229,607 and 496,822 effective tags were obtained for 16S and ITS sequencing, respectively, under different heavy metal stress conditions. The good coverage of all the samples was higher than 0.999, indicating the sequencing results obtained a sufficiently reliable data quality. All the rarefaction curves of the samples tended to be smooth as the reads increased, indicating there was enough data for each sample and sufficient deep sequencing (Figure 1) [15].

The term “alpha diversity” refers to the level of diversity within a specific geographical region or ecosystem. The Shannon and Simpson indices are commonly used to quantify species diversity. A higher value of the Shannon index indicates greater microbial community diversity in the sample. When exposed to different heavy metal stressors, fungal communities exhibited the highest richness and diversity under Pb stress, followed by Zn and Cd stress. Similarly, the bacterial communities showed the highest richness and diversity under Zn stress, followed by Cd and Pb stress (Table 1). These findings suggest that endophytic fungi and bacteria associated with comfrey plants are influenced by heavy metals, such as Pb and Zn.

The diversity of fungal communities showed that the number of species and Shannon index of fungi in heavy metal-contaminated host plants were lower than those in uncontaminated plants [17]. These indicators were the lowest in contaminated plants, indicating that endophytic fungi were unable to resist the external environment when the host plant faced abiotic stress, resulting in a decrease in the number and diversity of flora. The α diversity index of the endophytic bacteria was higher than that of the control group, suggesting that the endophytic bacteria resist the external environment by increasing the number and diversity of flora when the host plant is subjected to abiotic stress. These results indicate that heavy metals inhibited the diversity and richness of fungi but enhanced the diversity and richness of bacteria.

### 3.3. Composition of the Fungal and Bacterial Communities

Based on the taxonomic affiliations of the operational taxonomic units (OTUs), the analysis revealed the presence of 37 phyla (97.30% classified), 78 classes (92.30% classified), 175 orders (91.43% classified), 329 families (89.36% classified), and 614 genera (87.30% classified) under various heavy metal stress conditions. The relative community abundance of the top ten fungal phyla at the phylum level is visually represented in Figure 2.

In the root samples of CK, Zn, Cd, and Pb, the dominant fungal phylum was Ascomycota, accounting for relative abundances of 63.45%, 50.92%, 67.09%, and 68.38%, respectively (Figure 2A). At the genus level, Dactylaria emerged as the dominant genus in the CK, Zn, and Cd root samples, with relative abundances of 20.99%, 20.89%, and 22.14%, respectively. Cladophialophora was identified as the dominant genus in the Pb samples, constituting 17.32% (Figure 2B). In the context of different heavy metal stress conditions, the bacterial OTUs were classified into 15 phyla and 572 genera. Proteobacteria dominated the bacterial phylum in the CK, Cd, Pb, and Zn samples, with relative abundances of 40.32%, 69.42%, 57.92%, and 69.79%, respectively (Figure 2C). At the genus level, Chloroplast was identified as the dominant genus in the CK samples, accounting for 22.71%. Cellvibrio emerged as the dominant genus in the Cd stress samples, constituting 29.27%, while Flavobacterium dominated in the Pb and Zn stress samples, with relative abundances of 25.55% and 18.11%, respectively (Figure 2D).

### 3.4. Shifts of the Bacterial and Fungal Community Structures under the Different Heavy Metal Stress Conditions

A PCoA was used to compare the bacterial community composition at the OTU level under three heavy metal stress conditions. It was used to further explore the differences in the community compositions between different heavy metal stress conditions. This PCoA demonstrated that heavy metal stress causes significant differences in the endophyte communities of *S. officinale*.

PCoA1 and PCoA2 accounted for 40.6% and 20.8% of the observed variation, respectively. The endophytic bacteria under different heavy metal stress conditions exhibited clear segregation into distinct groups, with the endophytic bacteria under Pb stress exerting a pronounced influence on the observed variation. The SPERMANOVA analysis, based on the weighted UniFrac distances, further validates that the endophytic bacteria under Pb stress significantly impact the composition of the bacterial communities (Figure 3A). There was no significant difference in the composition of the endophytic fungi under different heavy metal stress conditions (Figure 3B).

### 3.5. Heavy Metal Stress-Associated Endophyte Biomarkers and Specific OTUs

The Venn diagram illustrates a cluster representing the shared operational taxonomic units (OTUs) and distinctive OTUs for each sample [4]. A total of 4006 endophytic bacterial OTUs were annotated based on the classification level of 97% similarity. Among them, the results showed that there were 1004 specific OTUs under Cd stress, including 23 phyla and 206 genera, among which Proteobacteria (62.8%) was the dominant phyla. *Pseudomonas* (2.6%) was the dominant genera. There were 597 specific OTUs under Pb stress, including 17 phyla and 127 genera. Among them, Proteobacteria (62.0%) was the dominant phyla. *Flavobacterium* (5.8%) was the dominant genera. Under Zn stress, there were 1244 specific OTUs, including 20 phyla and 240 genera, among which Proteobacteria (57.4%) was the dominant phyla. *Allorhizobium-Neorhizobium-Pararhizobium-Rhizobium* (5.01%) was the dominant genera. There were 531 specific OTUs in the control group, including 19 phyla and 156 genera. Among them, Proteobacteria (40.0%) was the dominant phyla. *Pseudomonas* (10%) was the dominant genera. Compared with the control group, Proteobacteria was still the dominant phylum at the gate level after heavy metal stress in the host plant, and the proportion of Proteobacteria was significantly increased, indicating that the bacteria belonging to Proteobacteria resist the adverse external environment by increasing the number of bacteria when the host plant is faced with abiotic stress, which plays an important role in helping the host plant reduce tolerance to heavy metals. At the genera level, the specificity of the bacterium *Pseudomonas* was under Cb and Pb stress. *Flavobacterium* is a specific bacterium. The *Flavobacterium* and *Allorhizobium—Neorhizobium—Pararhizobium* specificity of the Rhizobium bacteria genera have certain specificity at the genus level (Figure 4A).

A total of 728 endophytic fungal OTU sequences were analyzed, and the results showed that there were 110 specific OTUs under Cd stress, including 8 phyla and 33 genera, among which Ascomycota (27.5%) was the dominant phyla. *Exophiala* (2.7%) was the dominant genera. There were 81 specific OTUs under Pb stress, including 6 phyla and 33 genera, among which Ascomycota (40.6%) was the dominant phyla. *Cladophialophora* (3.7%) was the dominant genera. There were 135 specific OTUs under Zn stress, including 6 phyla and 45 genera, among which Ascomycota (35.8%) was the dominant phyla. *Flagelloscypha* (2.9%) was the dominant genera. There were 257 specific OTUs in the control group, including 5 phyla and 88 genera, of which Ascomycota (44.9%) was the dominant phyla. *Comoclathris* (3.5%) was the dominant genera. In conclusion, Ascomycota was the dominant phylum in all the groups, and compared with the control group, the proportion of Ascomycota in the heavy metal treatment group was reduced, indicating that the fungi belonging to Ascomycota had weaker resistance to external adverse environments when the host plants were faced with abiotic stress. At the genus level, endophytic fungi also have certain specificity under different heavy metal stress conditions. For example, *Exophiala* is a specific genus under Cd stress, and the same is true for *Cladophialophora* under Pb stress and *Flagelloscypha* under Zn stress (Figure 4B).

The LEfSe analysis revealed differences in the bacterial and fungal communities under heavy metal stress. In order to detect the endophyte taxonomic biomarkers of the fungal and bacterial communities, OTUs exhibiting a relative abundance greater than 0.05% were compared among the samples utilizing LEfSe (linear discriminant analysis effect size) [18]. The OTUs with the highest LDA (Wilcoxon *p* < 0.05, LDA score > 3) from each endophyte are visually presented in Figure 4D,F. Higher LDA scores represent a greater effect of species abundance on variation. Microorganisms with an LDA > 3.5 and *p* < 0.05 were screened as microbial biomarkers [19]. In the fungal community, a total of twenty fungal clades were selected, including three phyla, seven classes, five orders, five families, and four genera. Further, LDA was used to compare the effect of each fungal community abundance on each heavy metal [20]. It can be seen in Figure 4E that *Agaricomycetes Chaetasbolisia_eupatorii*, *Pleosporales Chaetasbolisia*, and *Didymellaceae* were the biomarkers in the fungi under the Zn stress condition. However, in fungi under the Pd stress condition, *Microbotryomycetes* was identified as a biomarker. However, in the bacterial community, a total of six bacterial clades were selected, including three phyla, three classes, three orders, three families, and two genera. It can be seen in Figure 4C that Bdellovibrionia, Bacteriovoracaceae, Actinoallomurus, and Bacteriovoracales were the biomarkers in the bacteria under the Zn stress condition, while in the bacteria, under Cd stress, Peredibacter and Oceanospirillales were identified as the biomarkers. This was roughly the same as the results of the analysis of the dominant phylum and dominant genus in Figure 2. The results rendered it more evident that heavy metals affect the species of dominant bacterial communities in endophytes [21].

### 3.6. Co-Occurrence Network Analysis of the Fungal and Bacterial Community

In cases where the relative abundance of different entities under heavy metal stress exceeded 0.02% ASV, we constructed a network diagram to depict the co-occurrence patterns within the bacterial and fungal communities [6]. Subsequent analysis of the asymbiotic network revealed that, on average, the bacteria exhibited a higher degree and graph density, accompanied by a shorter average network path length(Figure 5A,C,E,G). This suggests a closer and more interactive relationship among the bacteria. Notably, modular interactions among the endophytic bacteria were observed under Zn and Pb stress conditions(Figure 5C,G), signifying the evolution of distinct interaction relationships to withstand adverse external conditions. In contrast, compared to the control group, interactions among the endophytic fungi were diminished, indicating that the endophytic fungi exhibited reduced resilience to abiotic stresses within the host plants(Figure 5B,D,F,H). Meanwhile, the endophytic bacteria demonstrated an ability to resist adverse external environments by enhancing community interactions and altering the interaction patterns. Overall, these findings underscore the greater resilience of endophytic bacteria to heavy metal stress compared to endophytic fungi [22].

To explore the impact of heavy metals (Pb, Zn, and Cd) on the pepper microbiome co-occurrence patterns, we conducted an analysis of the bacterial–bacterial and fungal–fungal intrakingdom networks, along with the bacterial–fungal interkingdom networks [23]. In the intrakingdom network analysis, we observed a greater proportion of negative edges and higher modularity in the bacterial networks compared to the fungal networks. Furthermore, the bacterial networks exhibited a higher number of nodes and edges when compared to the fungal networks. Additionally, the bacterial network under heavy metal stress demonstrated increased complexity, as indicated by the higher number of nodes and edges. Specifically, the bacterial taxa showed higher network connectivity in the microbial networks than the fungal taxa. These results indicated that different heavy metal stress conditions enhanced the bacteria–bacteria intrakingdom associations but decreased the fungi–fungi associations in *S. officinale*.

### 3.7. Environmental Factors Correlated with Heavy Metal Contamination

The Spearman correlation analysis showed that the α diversity index of the endophytic fungi (chao1, simpson, and shannon) was negatively correlated with the transport coefficient, bioconcentration coefficient, and retention rate of the three heavy metals, and the chao1 index was significantly negatively correlated with the allocation coefficient. The α diversity index of the endophytic bacteria was positively correlated with its transport coefficient, distribution coefficient, and retention rate, and the simpson index was significantly positively correlated with the transport coefficient (Figure 6).

FUNGuild is a commonly utilized tool for predicting the nutritional and functional groups within fungal communities. A total of 31 functional pathways were analyzed for endophytic fungal communities, of which the most abundant were Unassigned, Plant_Pathogen, Undefined_Saprotroph, Lichen_Parasite, and Plant_pathogenic_(?)_on_polen. The results showed that functional groups, such as Plant_pathogenic_(?)_on_polen, Orchid_Mycorrhizal, Undefined_Saprotroph, Plant_Pathogen-Wood_Saprotroph, and Soil_Saprotroph had significant differences among the four samples. In particular, Plant_pathogenic_(?)_on_polen was more abundant in the CK sample, Soil_Saprotroph was more abundant under the Zn stress condition, Undefined_Saprotroph, and Plant_Pathogen-Wood_Saprotroph were more abundant under the Pb stress condition, and Orchid_Mycorrhizal was more abundant under the Cd stress condition. In the control group, plant pathogens were the main trophic modes, but when the host plants were subjected to abiotic stress, the trophic modes of the heavy metal treatment group were different, indicating that heavy metal stress can change the trophic modes of the host plants, and the trophic modes have certain specificity for different heavy metals. Furthermore, the trophic modes changed from plant pathogens to saprophytic bacteria. This shows that saprophytic bacteria can adapt to heavy metal stress better.

To investigate bacterial function, we conducted a bacterial function prediction analysis using PICRUSt. By comparing the results with the Kyoto encyclopedia of genes and genomes (KEGGs) database, the PICRUSt analysis of the bacterial 16S rDNA sequences consistently revealed “Membrane_Transport” as the predominant function across all the samples. Compared with the control group, the proportion of Membrane_Transpor increased in the heavy metal treatment group, indicating that the endophytic bacteria helped host plants resist the Membrane_Transport function in the face of abiotic stress. In addition, a significantly enhanced function in the heavy metal treatment group was Energy_Metabolism, which provides energy for the resistance of host plants.

The results show that the trophic mode of endophytic fungi differed in different heavy metal stress conditions. Plant_Pathogen was the dominant trophic mode in the Zn and Cd samples (23.57% and 22.59%), while Undefined_Saprotroph was the dominant trophic mode in the Pb samples (46.69%) (Figure 7).

To investigate bacterial function, we employed PICRUSt for the bacterial function prediction analysis. Upon comparison with the Kyoto encyclopedia of genes and genomes (KEGGs) database, the PICRUSt analysis of the bacterial 16S rDNA sequences consistently identified “Membrane_Transport” as the main function in all the samples, constituting a substantial proportion ranging from 10.94% to 13.82% (Figure 7).

## 4. Discussion

Numerous studies have emphasized the impact of host species, tissue types, and plant growth environment on the diversity of endophytes. Co-operative and competitive interactions among microbial species, as well as network modularity, play pivotal roles in influencing community stability [24], as reported in previous research [25]. In the current study, bacterial networks and their hub taxa in both control (CK) and treated samples exhibited a higher proportion of negative correlations compared to the fungal networks. The prevalence of mutually negative interactions, indicative of ecological competition, contributes to microbiome stability by mitigating the destabilizing effects of co-operation. The host may derive benefits from microbial competition, leading to enhanced resistance to external stress. In contrast to bacterial communities, fungal communities were more significantly affected by heavy metals, likely due to the intensified positive intrakingdom correlations among the fungal taxa observed in the networks compared to the healthy networks. Additionally, the lower modularity in the fungal network may exacerbate destabilization due to the higher prevalence of cross-module correlations among taxa. These findings suggest that fungal communities are more sensitive to heavy metals than bacterial communities, as evidenced by their lower network stability [26]. A prior study indicated that soil bacterial networks remained stable under drought stress compared to fungal networks. Furthermore, specific bacterial taxa, such as Proteobacteria, Actinobacteria, and Myxococcota, enriched in diseased plants, were identified as hub taxa in the co-occurrence networks. Hub taxa, occupying key topological positions within the network, may play a crucial role in orchestrating favorable plant microbiomes.

The results showed that the effect of heavy metals on bacterial communities was greater than that on fungal communities. The differences between the bacteria and fungi in body size, living habits, metabolic activity, diffusion potential, and host-microbial interactions may affect the process of species sorting and community assembly.

The data suggest that heavy metal stress increases the complexity of bacterial networks, thereby increasing the host plant’s tolerance to heavy metals. Previous research has shown the complexity of networks and the importance of key class groups in supporting ecosystem functions [27].

Many studies have consistently reported that bacterial and fungal communities in plants, analyzed through high-throughput sequencing (HTS) methods, exhibit a few dominant phyla [7]. For bacteria, these commonly include Proteobacteria, Bacteroidetes, and Actinobacteria, while fungi often showcase dominance with Ascomycota, Basidiomycota, and Zygomycota. In the case of *S. officinale*, under different heavy metal stress conditions in this study, the endophytic bacterial and fungal communities were categorized into 27 and 13 phyla, respectively. The prevailing phyla were Proteobacteria for bacteria and Ascomycetes for fungi, aligning with the patterns observed in prior studies. It is noteworthy that although Proteobacteria and Ascomycetes remained dominant under all heavy metal stress conditions, their relative abundances varied. However, there were significant differences in the abundance of fungal and bacterial two-point dominant strains under different heavy metal stress conditions, which indicated that the bacterial and fungal communities had certain specificity [28].

However, there was some evidence that different metals affect the composition of endophytic communities to varying degrees. Cd, Zn, and Pb ions had significant effects on the diversity and species richness of the endophytic bacterial communities and changed their community composition. In fungal communities, Cd reduced the abundance of the dominant bacterial group, Ascomycota, while in the endogenic bacteria, Pb significantly reduced the abundance of Pseudomonas in the bacterial communities. In contrast, long-term Cu exposure was shown to change the bacterial community composition but not its diversity or species richness [29]. The results of other studies remain inconclusive because the effects of heavy metals are difficult to separate from the effects of other environmental factors [30].

Many studies have reported that the diversity of endophytes is affected by the environment of the host plant, especially under heavy metal stress. Due to different types of heavy metals, the diversity and richness of endophyte diversity significantly change [25]. In addition, the alpha diversity of endophyte communities decreased, while the richness increased under high concentrations of heavy metals in *R. pseudoacacia*. When Krishnamoorthy observed the effects of different heavy metal concentrations on the community structure, the abundance and species richness of the soil ambushed mycorrhiza communities, and the microbial richness and diversity in the Glomeraceae family increased correspondingly with the application of Pb, Cd, and Zn concentrations in soil. Compared with the control group, the diversity and abundance of the endogenic bacteria decreased after heavy metal stress. In endogenic bacteria, the diversity and abundance of endophytic fungi in the heavy metal treatment group were increased compared with that in the light group. We speculated that plant species influenced the selection of endophytic bacteria. The results showed that there were differences in the number of fungi and bacteria in plants under different stress environments. However, because bacteria and fungi greatly differ in size and life cycles, we speculated that they might respond differently to the living environment [31].

The results showed that Ascomycota and Proteobacteria were the keystone taxa of fungi and bacteria. Respectively, many studies have found that Proteobacteria is a common endophyte of many heavy metal-tolerant plants (such as *Salix atrocinerea* and *Solanum nigrum* L.), which may play a key role in the survival of hostile environments. Indeed, it was reported that Pseudomonas species are capable of synthesizing plant growth-promoting (PGP) substances, such as phytohormones, to enhance plant growth. Additionally, some Pseudomonas strains exhibit the ability to produce antibiotics, contributing to increased resistance against pathogens in plants. This dual capability renders Pseudomonas a valuable player in promoting both the growth and defense mechanisms of plants. Based on the heat map generated with the correlation coefficients, we conclude that fungi mainly play a role in membrane transporters and energy metabolism, and that fungi may enhance membrane transporters to move heavy metal ions, thereby protecting the host plant from heavy metals. The nutrient mode of bacteria changed from plant pathogenic bacteria to the saprophytic type, indicating that the bacteria may better survive by changing their nutrient mode when host plants are subjected to heavy metal stress, to ensure the normal growth and development of host plants. Endophytic bacteria and endophytic fungi, as the key groups of heavy metal stress, showed different responses, which not only contributed to the uptake of nutrients, but also resisted the persistent existence of heavy metals on cells. Secondly, different books of *S. officinale* of the same phylum have different correlations with environmental factors, indicating that the role of endophytes of the same genus is similar in different modules, but the function of each module is different [32].

The visually microbial symbiotic network visually the relationships among the endophytic flora and how they work to help the host plant survive in an unfavorable environment, with the heterogeneity of topological parameters in the endophytic bacterial network being higher than that in the endophytic fungal network [33]. As the initial compartment to combat heavy metal stress, roots prompted the endophytes to develop a more intricate division of labor, as shown by the increasing number of nodes, links, and modularity, indicating multiple living strategies and functional modules. In line with previous findings, the symbiotic relationships of the endophytic community of *S. officinale* shown in these results also represent a high variability in response to different environments, meaning more complex networks. It also suggested that network stability was significantly affected by heavy metal content. Similar to our findings, to maintain the normal physiological activities of roots and the functional stability of the endophytic community, more adaptive community division and reproduction were promoted, and multiple functional modules were formed to jointly maintain the ecological balance of the root environment. Heavy metal contamination may diversify microbial survival strategies for niche and nutrient acquisition, leading to module variations in keystone taxa. In summary, the difference in heavy metals is one of the main reasons for the differences in the symbiotic network of endophyte communities. Compared with the control group, with the application of heavy metals, the material exchange and signal exchange between the network species decreased, the stability of the community composition worsened, and the symbiotic relationship between microorganisms significantly decreased. In this regard, species in the same module may share similar niche preferences and functional roles, and heavy metal contamination may diversify microbial survival strategies for niche and nutrient acquisition, leading to module variations in keystone taxa. 

## 5. Conclusions

In summary, the study revealed notable changes in the community structure and diversity of endophytic fungi and bacteria under different heavy metal stress conditions. The overall trend observed was an increase in the community structure and diversity of endophytic bacteria, contrasting with a decrease in these parameters for endophytic fungi. The impact of different heavy metal stress conditions was evident in the altered dominance of species and ecological functions among the endophytes. The biomarker and specific OTU analyses highlighted distinct endophytes responding to specific heavy metal stress conditions.

Notably, endophytic bacterial taxa displayed higher network connectivity in microbial networks compared to fungal taxa. The Spearman analysis indicated a significant positive correlation between the diversity of endophytic bacteria and heavy metal transport. These findings contribute to a deeper understanding of plant–microbial interactions and offer valuable insights into the role of endophytes under varying heavy metal stress conditions.

In light of the results, it is inferred that endophytic bacteria in *S. officinale* exhibit enhanced resistance to adverse external environments, aiding the host plant in maintaining normal growth under both biological and abiotic stress conditions. The co-occurrence network diagram visually illustrates that under stressful conditions, bacterial interactions are closer, and a modular connection relationship evolves to provide protective benefits for the host plant. This provides a novel perspective on the function of endophytic bacteria, suggesting that they have significant potential to alleviate heavy metal stress. Future exploration through metagenomics may further elucidate the mechanisms underlying these interactions and responses.

## Figures and Tables

**Figure 1 microorganisms-12-00477-f001:**
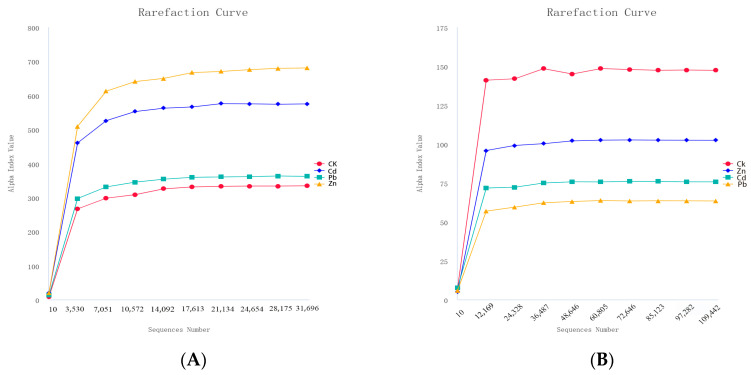
Rarefaction curves of bacteria and fungi under different heavy metal stress conditions ((**A**) bacteria, (**B**) fungi).

**Figure 2 microorganisms-12-00477-f002:**
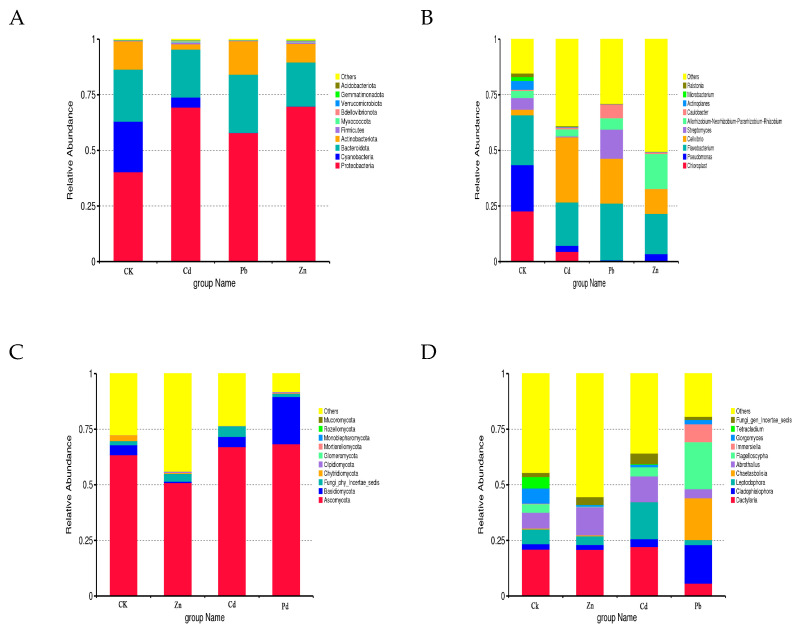
Bacterial and fungal community composition at the phylum and genus level. ((**A**) Bacterial community at the phylum level, (**B**) bacterial community at the genus level, (**C**) fungal community composition at the phylum level, (**D**) fungal community composition at the genus level).

**Figure 3 microorganisms-12-00477-f003:**
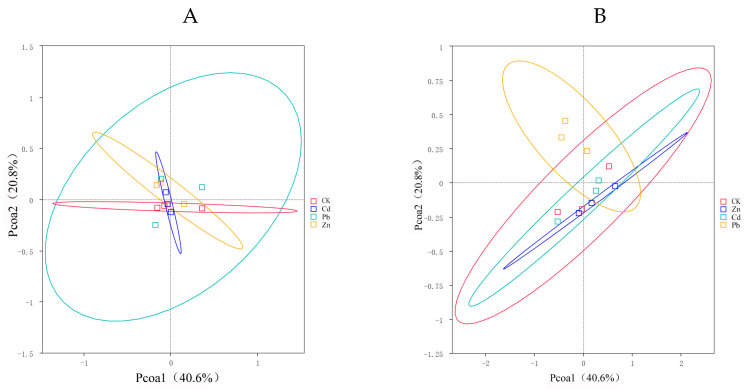
PCoA (principal coordinates analysis) serves as a technique for extracting the major variables from the intricate multidimensional variable data and presenting them visually ((**A**) bacteria, (**B**) fungi).

**Figure 4 microorganisms-12-00477-f004:**
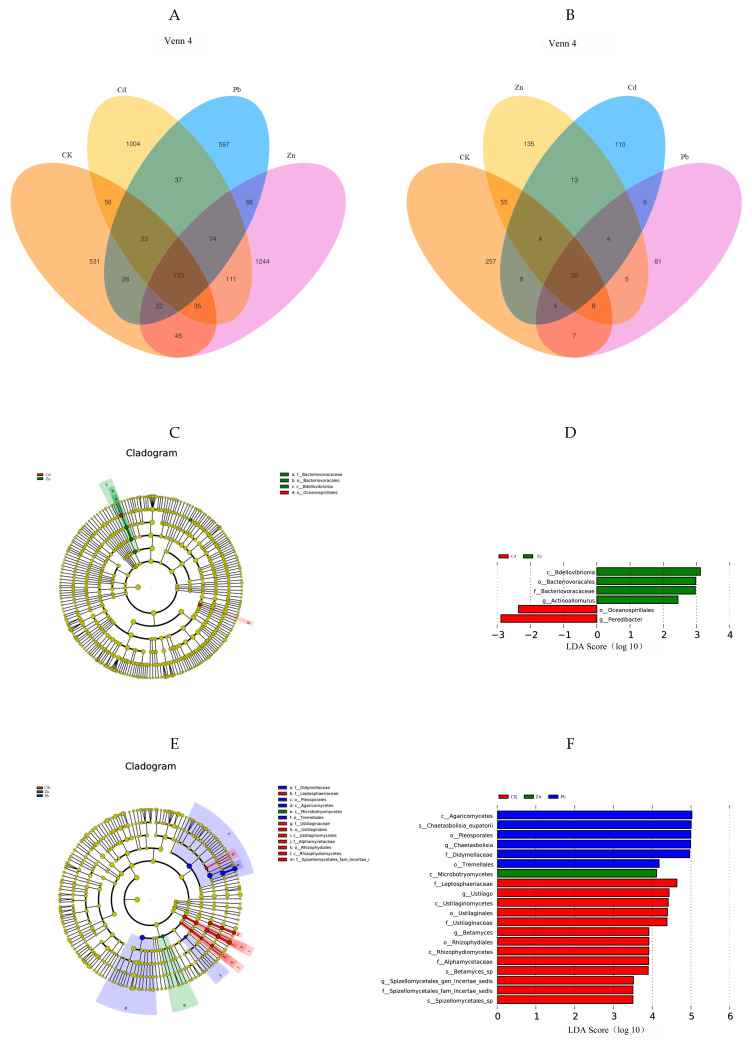
Venn diagrams showing the shared and specific OTUs among different heavy metal stress conditions. ((**A**) bacteria, (**B**) fungi). In the LDA (linear discriminant analysis) tree diagram, the concentric circles extending from the center outward depict the taxonomic levels ranging from phylum to genus (or species). Each small circle at a distinct classification level represents a classification at that specific level, and the diameter of each small circle is proportional to the relative abundance size ((**C**) bacteria, (**E**) fungi). The size of the LDA score signifies the influence of various species, with those having an LDA score exceeding 2 considered as confidently distinctive species. A higher LDA score value indicates a more substantial impact of the respective different species ((**D**) bacteria, (**F**) fungi).

**Figure 5 microorganisms-12-00477-f005:**
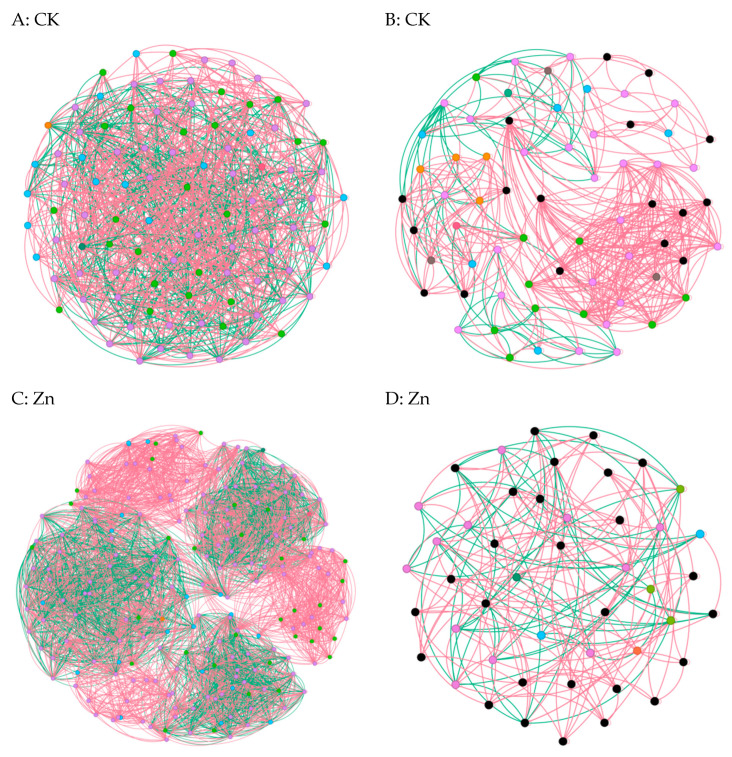
Co-occurrence network diagrams of the samples among the bacterial and fungal communities (bacterial (**A**,**C**,**E**,**G**); fungi (**B**,**D**,**F**,**H**)). The green and red lines represent co-occurrence and mutual exclusion, respectively.

**Figure 6 microorganisms-12-00477-f006:**
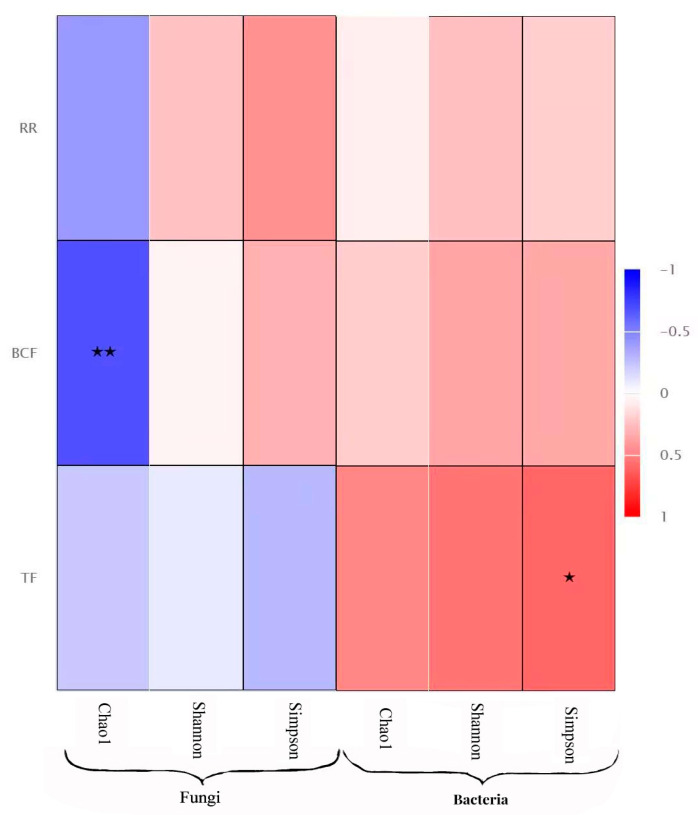
Correlation analysis between Environmental factors (transport coefficient, bioconcentration coefficient and retention rate of the heavy metals) and diversity of endophytes. Note: * indicate the differences are significant at *p* < 0.05 and ** indicate the differences are significant at *p* < 0.01.

**Figure 7 microorganisms-12-00477-f007:**
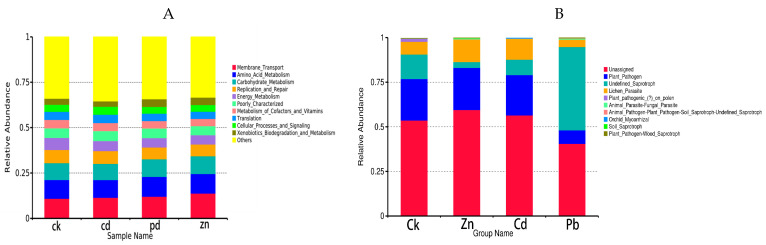
Predicted gene functional categories of the endophytic microbiome among different heavy metal stress conditions, The heat map is drawn for a total of 10 metabolic pathways, which were significantly changed in the bacterial and fungal communities ((**A**) bacteria, (**B**) fungi).

**Table 1 microorganisms-12-00477-t001:** Alpha diversity of the bacterial and fungal communities and sample coverage.

Sample	Endophytic Fungi				Endophytic Bacteria			
	Chao1	Shannon	Simpson	Goods Coverage	Chao1	Shannon	Simpson	Goods Coverage
CK	147.023	3.251	0.810	0.999	342.410	3.519	0.701	0.999
Cd	64.149	2.745	0.787	0.999	568.920	5.389	0.889	0.999
Pb	103.674	2.746	0.723	0.999	361.742	4.081	0.794	0.999
Zn	75.334	2.797	0.742	0.999	686.441	6.111	0.954	0.999

## Data Availability

The 16S rRNA and ITS gene sequences of endophytes used in this manuscript were submitted to the NCBI and the Accession number is NCBI: PRJNA1054967 (https://www.ncbi.nlm.nih.gov/ accessed on 15 December 2023)The 16S rRNA and ITS gene sequences of endophytes used in this manuscript were submitted to the NCBI and the Accession number is NCBI: PRJNA1054967 (https://www.ncbi.nlm.nih.gov/ accessed on 15 December 2023).

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
