# Peer review of "Effects of Different Heavy Metal Stressors on the Endophytic Community Composition and Diversity of Symphytum officinale"

_microorganisms, 2024, doi:10.3390/microorganisms12030477_

Round 1

Reviewer 1 Report

Comments and Suggestions for Authors

Please add to material and methods altitude for plant picking.

Please add to the materials and method of the heavy metal producer or company where they were purchased.

How were primers designed or purchased?

There are many formal deficiencies in the work, especially in citing sources, and others that need to be corrected.

Figure 4, 5, 6, 7 is illegible, they must be replaced.

Diskusia by mala byÅ¥ prepracované a porovnané z relevantnými údajmi.

The conclusion of the thesis must be rewritten with the most important findings.

Reviewer 2 Report

Comments and Suggestions for Authors

Dear Authors, please, first of all, adapt your works to the requirements of the journal!!! Overall, the aim of the work and the scope of the results obtained are interesting and worth publishing in Microorganisms . However, the manuscript requires major changes:

- the introduction should be expanded,

- adapt the literature references in the text and their list to the requirements of the journal,

- most figs. are unreadable,

- the methodology must be described in detail, e.g. accuracy of devices, etc.,

- please show the quality of the results obtained by preparing the manuscript.

Round 2

Reviewer 1 Report

Comments and Suggestions for Authors

Thanks to the authors for correction. Only small formal mistakes as space before parentheses, or Error! Reference source not found.

Reviewer 2 Report

Comments and Suggestions for Authors

The authors did a lot of work on the manuscript. However, still not all of its elements are optimally developed. In summary, the current version of the manuscript is significantly improved compared to the original version. However, some Figs (1-4 and 7) still require improvement and attention, which are difficult to read, and please pay attention to the notation of Latin names (we use the full name only when first using it in the text and then we use its abbreviation, e.g. see regarding the name Symphytum officinale).
